# OpenReview forum: "Do Vision & Language Decoders use Images and Text equally? How Self-consistent are their Explanations?"
_ICLR.cc/2025/Conference — ICLR 2025 Poster_

### Official Review · Reviewer_DVfk · 2024-10-26

**Soundness:** 3
**Presentation:** 4
**Contribution:** 4
**Rating:** 8
**Confidence:** 4

**Summary:**

The paper studies the degree to which VLMs use image or text modalities more in different contexts. The paper does a good job in extending established measures like MM-SHAP to decoder models and evaluating self-consistency of decoders in post-hoc and CoT with CC-SHAP. The authors do an extensive evaluation across 12 datasets with multiple answer formats.

The papers main contribution include:
1) Benchmarks VLMs on VALSE.
2) VL decoders are more text-centric compared to VL encoders.
3) VLMs are less self-consistent than LLMs.

**Strengths:**

1) Extending proven measures for VL Encoders to Decoders.
2) Provides robust evidence about how VLMs rely on different modalities unequally, by evaluating extensively across 12 datasets.

**Weaknesses:**

1) The paper evaluates only on LLaVA based open-source models. Could the authors evaluate on mPLUG-Owl2 or CogVLM or other non-LLaVA based VLMs as it would be ideal to see the approach work on a different architecture and with different pre-training data? [RESOLVED]

**Questions:**

Please refer to the weakness section to answer questions about the paper.

---

> ### Author Response · Authors · 2024-11-17
> **Response to initial review**
>
> Thank you for your positive review. We are glad that you highlight our sound motivation and beneficial results, writing and extensive experimental results on many datasets. We address your concerns individually below.
>
> ### Experiments with New Model
> > Only evaluated on LLaVA-based models, missing baselines such as CogVLM.
>
> Yes, we acknowledge that all experiments in the original submission were conducted using LLaVA-based models. Following your valuable feedback (aligned with the suggestions made by reviewers DVfk and BrVx), we invested significant effort into implementing our tests and analyses for mPLUG-Owl3 (which is the reason for the delay in responding to this discussion). The results of these experiments are now included throughout the revised version of the paper (see Tables 1-3 and Figures 1, 4, and 5).
>
> The findings for mPLUG-Owl3 (see below for our reasons for selecting this model) align with the overarching conclusions of the paper with the previously tested models:
> * Predominant reliance of the tested VLMs on text
> * A statistically significant higher contribution of text when answering compared to explaining – to a degree closer to LLaVA-NeXT-Vicuna than to LLaVA-NeXT-Mistral or BakLLaVA (also Mistral-based).
> * This pattern is stronger in CoT settings, again to a similar degree to LLaVA-NeXT-Vicuna than to the other 2 models.
>
> We thank you for your valuable suggestion, as we indeed agree that including the results of these additional models makes our findings more insightful and the paper more comprehensive.
>
> **Motivation for Model Choice**: We selected mPLUG-Owl3 based on Rev DVfk's recommendation of mPLUG-Owl2, opting for the newest and architecturally distinct model from LLaVA. Implementing our analyses for mPLUG-Owl3 required significant effort in such a short time due to its different chat template, necessitating substantial implementation work – finding time for this work was challenging. Additionally, compute queuing times and limited access to hardware (Ampere GPUs) made us exclude other strong candidate models, such as CogVLM. With more time and resources more VLMs could be added, but we had to set practical limits. To support further exploration, we provided the code for the community in the zip file attached to the submission.
>
>
> ---
> We hope that our response and the revisions to the paper have adequately addressed your concern. In light of this, we kindly ask you to reconsider your rating, especially given that you have rated our paper as "excellent" across presentation, and contribution, and “good” for soundness. If you have any additional comments or questions, we would be happy to engage further.

---

> ### Author Response · Authors · 2024-11-19
> **Thanks for raising your score**
>
> Thank you so much for raising your score from 6 to 8. We're very happy that the additional experiments with mPLUG-Owl3 strengthened the paper and resolved your concern that our experiments were too limited in the scope of model types. Could we kindly ask you to update your review and mark the weakness you see as [RESOLVED] so the area chair can clearly see that this is no longer an issue? We really appreciate your support!

---

### Official Review · Reviewer_BrVx · 2024-11-03

**Soundness:** 3
**Presentation:** 3
**Contribution:** 3
**Rating:** 6
**Confidence:** 3

**Summary:**

This work aims to answer an important research question - Do VLM-generated responses (and self-explanations) rely more on images or text?"

By extending methods from VL and LLM research, the authors propose MM-SHAP to measure the contribution of individual tokens to model predictions in order to explore the multimodal degree of autoregressive VLM under different tasks. Their findings are quite interesting and inspiring for future VLM works.

**Strengths:**

- This work aims to explore an important research question that is well-motivated. The three findings presented are all very interesting.
- The approach is clearly presented and reasonable.
- Comprehensively experiments and analysis.

**Weaknesses:**

- This work claims the scope as “all decoder VLMs,” yet all experiments are based on llava-based models. I agree with the statement in line 108 that these models (e.g. llava, miniGPT-4, blip2, Otter, etc) share similar high-level ideas; however, do their subtle differences in design lead to changes in the results/conclusions?
- The discussion section lacks in-depth analysis and exploration of the root causes of the observation. In particular, each paragraph in Section 4.4 reveals some interesting phenomena (for example, for CoT ability, VLM is clearly worse than that of LLM in self-consistency) but lacks a deeper discussion of the underlying reason. That being said, this is still a good finding paper.

**Questions:**

As mentioned in the Weaknesses section, do you think/have found your conclusions can be applied to all decoder VLMs?

---

> ### Author Response · Authors · 2024-11-17
> **Response to initial review**
>
> Thank you for your positive review. We are glad that you recognized our strong motivation, interesting findings, clear writing, and comprehensive experimental work and analyses. Below, we address your concerns and question individually.
>
> ### Experiments with New Model
> > Only evaluated on LLaVA-based models, missing baselines such as CogVLM.
>
> Yes, we acknowledge that all experiments in the original submission were conducted using LLaVA-based models. Following your valuable feedback (aligned with the suggestions made by reviewers DVfk and BrVx), we invested significant effort into implementing our tests and analyses for mPLUG-Owl3 (which is the reason for the delay in responding to this discussion). The results of these experiments are now included throughout the revised version of the paper (see Tables 1-3 and Figures 1, 4, and 5). We also made sure to replace statements like “for all VLMs” with more precise language, such as “all tested VLMs,” to ensure accuracy.
>
> The findings for mPLUG-Owl3 (see below for our reasons for selecting this model) align with the overarching conclusions of the paper with the previously tested models:
> * Predominant reliance of the tested VLMs on text
> * A statistically significant higher contribution of text when answering compared to explaining – to a degree closer to LLaVA-NeXT-Vicuna than to LLaVA-NeXT-Mistral or BakLLaVA (also Mistral-based).
> * This pattern is stronger in CoT settings, again to a similar degree to LLaVA-NeXT-Vicuna than to the other 2 models.
>
> We thank you for your valuable suggestion, as we indeed agree that including the results of these additional models makes our findings more insightful and the paper more comprehensive.
>
> **Motivation for Model Choice**: We selected mPLUG-Owl3 based on Rev DVfk's recommendation of mPLUG-Owl2, opting for the newest and architecturally distinct model from LLaVA. Implementing our analyses for mPLUG-Owl3 required significant effort in such a short time due to its different chat template, necessitating substantial implementation work – finding time for this work was challenging. Additionally, compute queuing times and limited access to hardware (Ampere GPUs) made us exclude other strong candidate models, such as CogVLM. With more time and resources more VLMs could be added, but we had to set practical limits. To support further exploration, we provided the code for the community in the zip file attached to the submission.
>
>
> ### Discussion Section
> > The discussion section lacks in-depth analysis and exploration of the root causes of the observation.
>
> Thank you for finding our findings interesting. We agree that further exploration for the root causes of our observation would be welcome. Understanding the root causes requires new methodological development, which were beyond the scope and page limit of this paper – which provides this analysis and highlights these findings in the first place. We view our findings as a foundation for further exploration, and we hope they will inspire follow-up research to find the root of these issues and fix them. Our experiments with mPLUG-Owl3 and LLaVA-NeXT-Vicuna show that there are models and architectures which suffer less from them.
>
> ---
> We hope that our response and the revisions to the paper have adequately addressed your primary concerns. In light of this, we kindly ask you to reconsider your rating, especially given that you have rated our paper as "good" across all three individual dimensions: soundness, presentation, and contribution. If you have any additional comments or questions, we would be happy to engage further.

---

> ### Author Response · Authors · 2024-11-22
> **Kind reminder**
>
> Dear Reviewer,
>
> We wanted to send a friendly reminder regarding our response. We believe that our response and the revised version of the paper, **including additional experiments with mPLUG-Owl3 (a model not based on LLaVA**, address the points raised in your review.
>
> If you have any remaining questions or concerns, we’d greatly appreciate the opportunity to address them during the discussion period.
>
> Thank you, and best regards!

---

> > ### Comment · Reviewer_BrVx · 2024-11-25
> > **Response to additional experiments**
> >
> > The supplemental experiment was interesting, so I will stay with the positive rating.

---

> > > ### Author Response · Authors · 2024-11-25
> > > **Thanks for acknowledging our extra experiments**
> > >
> > > Thanks for taking a look at our new experiments with mPLUG-Owl3. They are meant to address 1 of the 2 weaknesses in your review. Could we kindly ask you to update your review and possibly your scores and mark the weakness you see as [RESOLVED] so the area chair can clearly see that this is no longer an issue? We really appreciate your support!

---

### Official Review · Reviewer_ETD8 · 2024-11-04

**Soundness:** 3
**Presentation:** 2
**Contribution:** 3
**Rating:** 6
**Confidence:** 2

**Summary:**

This paper investigates how Vision-Language Models use image and text modalities in generating predictions and explanations. It studies whether VLMs rely more on visual or textual inputs, and also studies to what extent these models are self-consistent when providing explanations. The paper found that generally, LLM is more reliable than VLMs and text contributes more in VL decoders compared to image. And the effect is even larger with CoT explanations included. The paper also provided the first  benchmarking of state-of-the-art VL decoders on the VALSE benchmark.

**Strengths:**

1. The paper identifies and addresses a gap of previous works. Previous works only studies encoder while this work provides a study of the decoder on their multi-modal degree and the consistency in their self-explanation.
2. The paper found that VLMs use text more than image, and the gap is larger with CoT included.
3. The paper provides an benchmarking of state-of-the-art VL decoders on VALSE, which has previously only focused on encoders.

**Weaknesses:**

1. The paper primarily uses existing techniques (e.g., MM-SHAP and CC-SHAP) and the main contribution is applying these metrics to VLM decoders. This adaptation is incremental.
2. The writing is unclear and needs to be improved. It would be useful to provide more details on the Shapley values and the computation.

**Questions:**

1. The results show that text is more significantly used compared to image for answer generation. Is it possible that this is a result of dataset biases or limitations in the MM-SHAP metric’s rather than image/text contributions?

---

> ### Author Response · Authors · 2024-11-17
> **Response to initial review**
>
> Thank you for your positive review. We are glad that you recognized the gap in previous work which our paper addresses, our findings and new results from VLM benchmarking. Below, we address your concerns and your question individually.
>
> ### Contribution
> > The paper primarily uses existing techniques (e.g., MM-SHAP and CC-SHAP) and the main contribution is applying these metrics to VLM decoders.
>
> Our paper contributes **novel findings, conclusions, and analyses on VL decoders**, which have not been previously explored as such. Earlier investigations focused solely on accuracy using non-diagnostic benchmarks, unlike our comprehensive tests with VALSE. The methodological novelty of our work lies in extending MM-SHAP, initially developed for VL encoders, to VL decoders. Coupled with our numerous and original findings and analyses, we believe these are significant and noteworthy contributions.
>
> ### Explanation of Shapley Values
> > The writing is unclear and needs to be improved. It would be useful to provide more details on the Shapley values and the computation.
>
> In the original submission, we had already included details about Shapley values in the Appendix (page 15) and referenced them in the main text. Now, we’ve also added the computation of CC-SHAP to the Appendix (page 16). Unfortunately, due to space constraints, this background information no longer fits within the main body. We’re open to any suggestions on how to improve this further.
>
> ### Image contributions
> > The results show that text is more significantly used compared to image for answer generation. Is it possible that this is a result of dataset biases or limitations in the MM-SHAP metric’s rather than image/text contributions?
>
> Dataset biases when pre-training VLMs may very well induce what is known as unimodal collapse -- and which has been shown by prior work [1] to be correctly reflected by MM-SHAP. This reliance on text of VL decoders is expected, as these models are fundamentally based on LLMs and steered through text. Additionally, specific n-grams such as “How many” or “A” or “B” in prompts like “Please answer with A or B” often carry significant importance, influencing the output more than the image does. This inherent text bias further reinforces their reliance on text as a primary modality.
>
> A symptom of unimodal collapse is that a VLM can answer a question about an image without giving the image to the model as input. Inspired by this, measures before MM-SHAP compared the model accuracy with and without the image. [1] showed that this measurement only works for cases in which the model answers correctly with the image-question input (as opposed to using the modality but not answering correctly) and showed that MM-SHAP is able to capture such cases (as well as the clear-cut ones) much better.
>
> MM-SHAP achieves this by directly analyzing output probability differences and by computing Shapley values for each modality, offering a more precise assessment of modality usage. When MM-SHAP indicates a high contribution from the text modality, it reflects how the model processes and relies on the input modalities (+/- the imperfections of SHAP or any other interpretability method).
>
> ---
> We hope that our response and the revisions to the paper have adequately addressed your concern. In light of this, we kindly ask you to reconsider your rating. If you have any additional comments or questions, we would be happy to engage further.
>
> As an additional note and update we want to inform you that: based on feedback from other reviewers we have included additional experiments on the mPLUG-Owl3 model, to extend the variety of VLM architectures, to corroborate our results. The resulting changes (and minor text updates to fit all into 10 pages) are reflected in the updated version of the paper and code.
>
> [1] “MM-SHAP: A Performance-agnostic Metric for Measuring Multimodal Contributions in Vision and Language Models & Tasks”, Parcalabescu & Frank, ACL 2023, https://aclanthology.org/2023.acl-long.223

---

> ### Author Response · Authors · 2024-11-22
> **Friendly reminder**
>
> Dear Reviewer,
>
> With the end of the discussion period approaching, we wanted to send a friendly reminder regarding our response. We believe that our response addresses the points raised in your review, so if you have any remaining questions or concerns, we’d greatly appreciate the opportunity to address them during the discussion period.
>
> Thank you, and best regards!

---

> > ### Author Response · Authors · 2024-11-25
> > **Again a reminder**
> >
> > Dear Reviewer,
> >
> > The discussion period is ending tomorrow, and we believe we have addressed the concerns you raised in your review. We would greatly appreciate your feedback before the discussion period closes and kindly ask you to reconsider your rating in light of our response. Thank you!

---

> > > ### Comment · Reviewer_ETD8 · 2024-12-03
> > >
> > > I would like to thank the author for the response. It addresses some of my concerns. I'd like to maintain my rating of weak accept.

---

### Official Review · Reviewer_76uQ · 2024-11-05

**Soundness:** 3
**Presentation:** 3
**Contribution:** 2
**Rating:** 6
**Confidence:** 3

**Summary:**

This paper conducts a study on how the inputs to an autoregressive vision-language model (VLM) influence the outputs. Mainly the paper compares the influence of textual and visual input tokens on the output answer and explanation, evaluates the faithfulness of explanations using controlled modifications of the input. The paper studies three open-weight models: BakLLaVA, LLaVA-Mistral, and LLaVA-Vicuna. The methods of the analyses are adapted from prior work on interpreting LMs and encoder VLMs. The results show, among other things, that textual inputs influence the answer more than visual inputs, but that the influence of visual inputs increases when looking at influence on generated explanations.

**Strengths:**

- Interesting findings regarding interpretability of VLM decoders. These are the main qualitative findings: text input is more influential than visual input for answer prediction, visual input is more influential for explanation generation than for answer prediction, the contributions from each modality are different in answer prediction and explanation generation
- Evaluated on 3 models and 3+ datasets

**Weaknesses:**

- Methods are essentially the same as those from prior work
- The argument that the edit-based tests do not provide consistent/meaningful results has dubious value given that flaws in these tests have already been exposed in the context of LMs. The paper does not give examples of prior work claiming that these tests are useful for VLMs.
- The paper does not explain how CC-Shap works (page 4) -- how are the values computed?

**Questions:**

- Lines 357-358 say: "The large difference between overall higher accr and lower acc results suggests that VL decoders rely on linguistic priors to solve VALSE" - isn't this also because acc is a harder metric? How do you tease those two issues apart?
- What do you view as the main takeaway(s) from the edit-based tests?

---

> ### Author Response · Authors · 2024-11-17
> **Response to initial review [1/2]**
>
> Thank you for your insightful review. We are glad that you recognized our interesting findings and extensive experimental work across multiple datasets. Below, we address your concerns and questions individually.
>
> ### Contribution
> > Methods are essentially the same as those from prior work
>
> Our paper contributes **novel findings, conclusions, and analyses on VL decoders**, which have not been previously explored as such. Earlier investigations focused solely on accuracy using non-diagnostic benchmarks, unlike our comprehensive tests with VALSE. The methodological novelty of our work lies in extending MM-SHAP, initially developed for VL encoders, to VL decoders. Coupled with our numerous and original findings and analyses, we believe these are significant and noteworthy contributions.
>
> ### Edit-based tests
> > The takeaways from the experiments with perturbed inputs seem unclear. Lines 471-472 seem to suggest that the results from the edit-based tests are hard to interpret.
>
> Indeed, the difficulty in interpreting edit-based tests is precisely the point there. Like the CC-SHAP paper [1] before us, we critique the edit-based self-consistency tests (self-claimed as measures of faithfulness): when tested on the same models and data, these tests give very inconsistent answers (ranging from 0\% to 100\% faithfulness scores), making their results hard to interpret. **The purpose of that section is to examine whether the findings and criticisms raised in [1] regarding edit-based tests for LLMs also apply to VLMs. And we find that they do.**
>
> The inconsistency stems from several sources of error, including: i) reliance on semantic evaluations (which cannot be automated without introducing errors, and are sometimes subjective as given in the example in L503 of whether “on the sidewalk”, and post-edit emits “city intersection” given the picture in Tab. 5 mean the same or not); and ii)  the use of edits or counterfactuals: The edit or counterfactual test admits a large number of (counterfactual) edits that are produced by a trial-and-error search algorithm (or a specifically trained model). This often leads to an artificially inflated test setting: the more tries, the more probable it is to find such an edit, and finding such edits deems the model unfaithful. Since the number of executed tries is a hyperparameter, there is of course a risk that by adjusting this hyperparameter, the test delivers a variety of faithfulness scores.
>
> We have now made this critique more explicit in our paper revision (L159-L192 of the related work section). We there discuss several issues with these tests, including their reliance on semantic evaluations – whereas CC-SHAP is free from such requirements.
>
> [1] “On Measuring Faithfulness or Self-consistency of Natural Language Explanations”, Parcalabescu & Frank, ACL 2024, https://aclanthology.org/2024.acl-long.329
>
> ### CC-SHAP
> > The paper does not explain how CC-Shap works (page 4) -- how are the values computed?
>
> Thanks for pointing this out. We did not include an explanation of CC-SHAP in the original submission as here we only apply it (with the corresponding implementation effort required to carry it over to VLMs, as it was coded only for LLMs before). We provided only a high-level overview in the Related Work section (L205 in the revised paper, L196 in the original submission), as the detailed methodology is already described in the original publication. To address this concern, we have now included the full details in the Appendix B.2 due to space constraints in the main text which we reference in the main text at L347.

---

> ### Author Response · Authors · 2024-11-17
> **Response to initial review [2/2]**
>
> ### Acc vs. acc_r
> > Lines 357-358 say: "The large difference between overall higher accr and lower acc results suggests that VL decoders rely on linguistic priors to solve VALSE" - isn't this also because acc is a harder metric? How do you tease those two issues apart?
>
> Both metrics are straightforward to measure (they count how many times the model answered correctly), but the difference lies in the information provided in the two different settings the metrics apply to. In the acc_r setting, the model has access to both sentences and can make a pairwise comparison. This allows it to rely on linguistic plausibility biases, such as recognizing that “6 cats” are less likely to appear than “2 cats”, without necessarily grounding its answer in the image. On a concrete example with oversimplified examples for brevity (exact prompts are in Appendix B.5), this looks like this:
>
> **To measure acc**, the model is given an image and a single sentence, which could either be a correct caption or a foil (an incorrect caption). The model doesn’t know which:
> - Is it true that “there are two cats in the image”? (A) Yes, or (B) no? The correct answer is: (
> The model answers A or B, and acc is calculated by checking whether the label is correct.
>
> **To measure acc_r**, the model receives an image and two sentences, which are the caption and the foil.
> - Here are two sentences: (A) There are 2 cats in the image. (B) There are 6 cats in the image. Which one fits the image best, is it (A) or (B)? The correct answer is: (
> The model answers A or B, and acc_r is calculated by verifying if the response matches the correct label.
>
> We hope to have made the relevant section clearer in the paper revision (L295-314).
>
> ---
> We hope that our response and the revisions to the paper have adequately addressed your concerns. In light of this, we kindly ask you to reconsider your rating. If you have any additional comments or questions, we would be happy to engage further.
>
> As an additional note and update we want to inform you that: based on feedback from other reviewers we have included additional experiments on the mPLUG-Owl3 model, to extend the variety of VLM architectures, to corroborate our results. The resulting changes (and minor text updates to fit all into 10 pages) are reflected in the updated version of the paper and code.

---

> ### Author Response · Authors · 2024-11-22
> **Gentle Reminder**
>
> Dear Reviewer,
>
> As we are halfway through the discussion period, we wanted to send a friendly reminder regarding our response. We believe that our response, along with the **revised version of the paper – including clarifications regarding the input-edit based tests and additional explanations of CC-SHAP** – address the points raised in your review.
>
> If you have any remaining questions or concerns, we’d greatly appreciate the opportunity to address them during the discussion period.
>
> Thank you, and best regards!

---

> > ### Author Response · Authors · 2024-11-25
> > **Again a reminder**
> >
> > Dear Reviewer,
> >
> > The discussion period is ending tomorrow, and we believe we have addressed the concerns you raised in your review in our response along with the revised version of the paper – including clarifications regarding the input-edit based tests and additional explanations of CC-SHAP.
> >
> > We would greatly appreciate your feedback before the discussion period closes and kindly ask you to reconsider your rating in light of our response. Thank you!

---

> > > ### Comment · Reviewer_76uQ · 2024-11-27
> > > **Reviewer reply**
> > >
> > > Thanks for your response. Since the response addressed the point about edit-based tests, I have increased my rating to a 6. But the contribution regarding edit-based tests still seems weak to me (and I have included this in my review). The argument seems to be that the edit-based tests do not provide a meaningful analysis of VLMs. This argument would be valuable if a contrary claim (i.e. that these tests are useful for analyzing VLMs) has been made in prior work. The paper does not seem to point to a claim like this and actually points to work showing that the tests are ineffective for LMs. This is akin to introducing a bad baseline for some task (after this baseline has already been shown to perform poorly on a related task), and claiming that the poor performance of the baseline is interesting.

---

> ### Author Response · Authors · 2024-11-27
> **Author reply**
>
> Dear Reviewer, Thank you very much for your response and for increasing your score, we truly appreciate it.
>
> Regarding the inclusion of edit-based tests, we completely understand your point and we would entirely accept it, if we wouldn't live in a research landscape where the work [1] demonstrating the ineffectiveness of edit-based tests for LMs is still relatively new and singular: Unfortunately, in the LM space, there are still many researchers who continue the line of edit-based tests without recognizing their flaws. We included these tests for VLMs to address a potential critique: that while we extended CC-SHAP to evaluate VLMs, we didn’t consider other widely used LM tests. So, we did this evaluation and found supporting arguments for the criticisms raised in [1] regarding edit-based tests, and that they are equally applicable to VLMs. Our goal was to temper the enthusiasm for this line of work, especially since it is extensible to VLMs -- and this takes up only a small part of the main body of this paper.
>
> Regarding your third point in the weakness you still mention in the review "The paper does not explain how CC-Shap works (page 4) -- how are the values computed?" we believe we have addressed it in the updated version of the paper. Could you kindly confirm if you are satisfied with the changes and update your review to reflect this, and if not, tell us how to improve the writing regarding this?
>
> Thank you once again for your thoughtful feedback.

---

### Official Review · Reviewer_Ypmz · 2024-11-13

**Soundness:** 3
**Presentation:** 3
**Contribution:** 3
**Rating:** 6
**Confidence:** 3

**Summary:**

This paper investigates how VLMs balance text and image data when generating answers and explanations, evaluating whether their reliance on each modality shifts depending on the task. Additionally, the authors measure the self-consistency of VLMs by comparing answers and explanations generated in both post-hoc and Chain-of-Thought settings. The results show that VLMs are less consistent than LLMs, with visual information playing a more significant role in explanations than in answer generation.

**Strengths:**

- This paper is well-motivated and shows interesting findings of VLM. I believe that the result is welcome and beneficial to the community.
- The paper is overall well-written.
- The experimental results are extensive, including various datasets (VQA, GQA, Foi1It, MSCOCO, VALSE) in a wide range of tasks.

**Weaknesses:**

- The main technical contribution is limited since the proposed method is heavily based on MM-SHAP and CC-SHAP but applies to VL decoders instead of encoders.
- Only evaluated on LLaVA-based models, missing baselines such as CogVLM.

**Questions:**

- Please refer to the weakness.

---

> ### Author Response · Authors · 2024-11-17
> **Response to initial review by Reviewer Ypmz**
>
> Thank you for your positive review. We are glad that you recognized our strong motivation, impactful results, clear writing, and extensive experimental work across multiple datasets. Below, we address your concerns individually.
>
> ### Experiments with New Model
> > Only evaluated on LLaVA-based models, missing baselines such as CogVLM.
>
> Yes, we acknowledge that all experiments in the original submission were conducted using LLaVA-based models. Following your valuable feedback (aligned with the suggestions made by reviewers DVfk and BrVx), we invested significant effort into implementing our tests and analyses for mPLUG-Owl3 (which is the reason for the delay in responding to this discussion). The results of these experiments are now included throughout the revised version of the paper (see Tables 1-3 and Figures 1, 4, and 5).
>
> The findings for mPLUG-Owl3 (see below for our reasons for selecting this model) align with the overarching conclusions of the paper with the previously tested models:
> * Predominant reliance of the tested VLMs on text
> * A statistically significant higher contribution of text when answering compared to explaining – to a degree closer to LLaVA-NeXT-Vicuna than to LLaVA-NeXT-Mistral or BakLLaVA (also Mistral-based).
> * This pattern is stronger in CoT settings, again to a similar degree to LLaVA-NeXT-Vicuna than to the other 2 models.
>
> We thank you for your valuable suggestion, as we indeed agree that including the results of these additional models makes our findings more insightful and the paper more comprehensive.
>
> **Motivation for Model Choice**: We selected mPLUG-Owl3 based on Rev DVfk's recommendation of mPLUG-Owl2, opting for the newest and architecturally distinct model from LLaVA. Implementing our analyses for mPLUG-Owl3 required significant effort in such a short time due to its different chat template, necessitating substantial implementation work – finding time for this work was challenging. Additionally, compute queuing times and limited access to hardware (Ampere GPUs) made us exclude other strong candidate models, such as CogVLM. With more time and resources more VLMs could be added, but we had to set practical limits. To support further exploration, we provided the code for the community in the zip file attached to the submission.
>
> ### Contribution
> > The main technical contribution is limited since the proposed method is heavily based on MM-SHAP and CC-SHAP but applies to VL decoders instead of encoders.
>
> Our paper contributes **novel findings, conclusions, and analyses on VL decoders**, which have not been previously explored as such. Earlier investigations focused solely on accuracy using non-diagnostic benchmarks, unlike our comprehensive tests with VALSE. The methodological novelty of our work lies in extending MM-SHAP, initially developed for VL encoders, to VL decoders. Coupled with our numerous and original findings and analyses, we believe these are significant and noteworthy contributions.
>
> ---
> We hope that our response and the revisions to the paper have adequately addressed your primary concerns. In light of this, we kindly ask you to reconsider your rating, especially given that you have rated our paper as "good" across all three individual dimensions: soundness, presentation, and contribution. If you have any additional comments or questions, we would be happy to engage further.

---

> ### Author Response · Authors · 2024-11-20
> **Gentle Reminder**
>
> Dear Reviewer,
>
> As we are halfway through the discussion period, we wanted to send a friendly reminder regarding our response. We believe that our response, along with the revised version of the paper, **including additional experiments with mPLUG-Owl3 (a model not based on LLaVA**, addresses the points raised in your review.
>
> If you have any remaining questions or concerns, we’d greatly appreciate the opportunity to address them during the discussion period.
>
> Thank you, and best regards!

---

> ### Author Response · Authors · 2024-11-25
> **Again a reminder**
>
> Dear Reviewer,
>
> The discussion period is ending tomorrow, and we believe we have addressed the concerns you raised in your review through additional experiments with mPLUG-Owl3. We consider the novel findings of this paper to be an important and notable contribution.
>
> We would greatly appreciate your feedback before the discussion period closes and kindly ask you to reconsider your rating in light of our response. Thank you!

---

> > ### Comment · Reviewer_Ypmz · 2024-12-01
> > **Reviewer reply**
> >
> > Thank you for your response. It addressed my concerns about the paper, and I appreciate the effort. While I still find the technical contribution somewhat weak, the overall contribution of this paper is clear. Therefore, I will maintain my positive rating.

---

### Meta-Review · Area_Chair_Gk75 · 2024-12-21

**Metareview:**

### Summary:
This paper investigates how VLMs use text and image modalities when generating predictions and explanations, evaluating their self-consistency in both post-hoc and CoT settings. The key findings are that VLMs rely more heavily on text than images for predictions, but images become more important for explanations. The work also provides benchmarking of VL decoders on VALSE.

### Strengths:
1. Novel and important investigation of modality usage in VLMs
> "Interesting findings regarding interpretability of VLM decoders. These are the main qualitative findings: text input is more influential than visual input for answer prediction, visual input is more influential for explanation generation than for answer prediction" - Reviewer 76uQ

2. Comprehensive empirical evaluation across multiple datasets and models
> "The experimental results are extensive, including various datasets (VQA, GQA, FoiIt, MSCOCO, VALSE) in a wide range of tasks" - Reviewer Ypmz

3. Clear methodology extending existing techniques to new settings
> "The paper identifies and addresses a gap of previous works. Previous works only studies encoder while this work provides a study of the decoder on their multi-modal degree and consistency in their self-explanation" - Reviewer ETD8

### Weaknesses:
1. Initially limited model coverage, though addressed during discussion
> "The paper evaluates only on LLaVA-based models, missing baselines such as CogVLM" - Reviewer Ypmz

2. Technical novelty could be stronger
> "The paper primarily uses existing techniques (e.g., MM-SHAP and CC-SHAP) and the main contribution is applying these metrics to VLM decoders" - Reviewer ETD8

3. Discussion depth could be improved
> "The discussion section lacks in-depth analysis and exploration of the root causes of the observation" - Reviewer BrVx


### Justification:
I recommend accepting this paper as the strengths clearly outweigh the weaknesses. The work provides important insights into how VLMs process different modalities, with strong empirical backing across multiple datasets. While the technical novelty is incremental, the findings are valuable for the community's understanding of VLMs. The authors demonstrated commitment to improvement during the discussion period by adding substantial new experiments and clarifications that addressed the main reviewer concerns.

**Additional Comments On Reviewer Discussion:**

The authors were highly responsive during the discussion period and made significant improvements:

1. Added experiments with mPLUG-Owl3 to address model diversity concerns
2. Clarified methodology around edit-based tests and CC-SHAP computation
3. Improved discussion of findings and implications

Notably, Reviewer 76uQ increased their rating based on the authors' responses and additional experiments. Reviewer ETD8 maintained their rating but somewhat acknowledged the improvements. All reviewers agreed that the core contributions remained valuable.

---

### Decision · Program_Chairs · 2025-01-22

Accept (Poster)